# Exploring pathways to compulsory detention and ways to prevent repeat compulsory detentions in England; clinician perspectives

Ariana Kular [1]*, Mary Birken [1], Lisa Wood [1,2], Jordan Parkinson [1,3], Theresa Bacarese-Hamilton[1], Louise Blakley [4], Chloe Hutchings-Hay [5], Patrick Nyikavaranda [1,6], Dilshard Alam[1], Raphael Ogbolu[3], Caroline Bendall[2], Lai Tang [2], Amy Nickson[3], Cathryn Revell[3], Henrietta Mbeah-Bankas[1], Lizzie Mitchell [1], Kathleen Lindsay Fraser[1], Valerie Christina White[1], Fiona Lobban[3,7], Brynmor Lloyd-Evans[1], Sonia Johnson[1,8]

**1** Division of Psychiatry, Faculty of Brain Sciences, University College London, London, United Kingdom, **2** North East London NHS Foundation Trust, London, United Kingdom, **3** Lancashire and South Cumbria NHS Foundation Trust, Lancashire, United Kingdom, **4** Southern Health NHS Foundation Trust, Hampshire, United Kingdom, **5** National Eating Disorders Service, South London and Maudsley NHS Foundation Trust, London, United Kingdom, **6** Department of Primary Care & Public Health, Brighton & Sussex Medical School, University of Sussex, Brighton, United Kingdom, **7** Spectrum Centre for Mental Health Research, Faculty of Health & Medicine, Lancaster University, Lancaster, United Kingdom, **8** North London NHS Foundation Trust, London, United Kingdom

* a.kular@ucl.ac.uk

## Abstract

Rates of compulsory detention in psychiatric hospitals have risen over several decades in England and some other higher income countries. This study explores clinicians' perspectives on how service users come to be compulsorily detained in psychiatric hospitals and their suggestions for reducing these detentions in the future. Semi-structured qualitative interviews were conducted with 23 clinicians working with individuals who have been compulsorily detained under the Mental Health Act in England. Interviews were carried out by telephone or videoconference. Data was analysed using template analysis, which involved developing a structured framework to organise and analyse data and to develop themes. Three major themes were identified, with multiple sub-themes (a) service user factors that increase the risk of compulsory detention, including high levels of risk, previous/underlying trauma, medication non-adherence, service user perceptions of their mental health state, dis-advantage and discrimination, and lack of stability and involvement from family and social networks; (b) service-level reasons for being detained, including lack of com-munication and continuity of care, historical inability to obtain trust and confidence from parts of population, clinician biases and assumptions, lack of resources, lack of treatment and care variety, and systemic/institutional barriers to engagement; and (c) potential pathways to reducing compulsory detention, including increasing care quality and patient/family level interventions, investing in services, offering choice

**Data availability statement:** The data supporting this study consist of full qualitative interview transcripts. These transcripts may contain potentially identifiable information, even after pseudonymisation, as participants sometimes refer to specific people, places, or events. Furthermore, the content includes highly sensitive personal information relating to participants' experiences of mental health services and involuntary detention. Releasing these transcripts publicly could compromise participant confidentiality and well-being. Data access is therefore restricted in accordance with the ethical approval granted by the University College London (UCL) Research Ethics Committee. These restrictions are in place to ensure the privacy and safety of participants. For data access inquiries, please contact the following independent institutional body: UCL Research Ethics Committee Email: ethics@ucl.ac.uk

**Funding:** This report describes independent research funded by the National Institute of Health and Care Research Policy Research Programme (Ref: NIHR20173). The views expressed are those of the authors and not necessarily of the NIHR or the Department of Health and Social Care Research. The funders have no role in the design of this study and will not have any role during its execution, data analyses, and interpretation, but are expected to provide assistance to any enquiry, audit, or investigation related to the funded work.

**Competing interests:** The authors have declared that no competing interests exist.

regarding medication, offering alternatives to detention, and improving discharge planning. Our study advances the literature by highlighting systemic, patient-level, and service-level factors perceived as driving practice. Better-resourced community services and care planning and strategies to address unconscious bias are identified as potential routes to reducing detentions. However, significant limitations are a preponderance of London-based psychiatrists in our sample, which may affect the generalisability of the findings to other roles and locations, and a lack of corroboration of perceived causality with more objective data.

## Introduction

Rates of compulsory detention in psychiatric hospitals under mental health legislation have risen over several decades in England and some other European countries, including Austria, France and Spain, for reasons that are not fully understood [1–4] despite a major review of evidence and practice informing the Independent Review of the Mental Health Act [5]. In England, national data from 2022-2023 reported 51,312 detentions under the Mental Health Act (MHA) (1983), the legal framework in England for detaining people in hospital [6].

Preventing detentions wherever possible is a national and global priority as detention is an inherently coercive action which does not appear to confer great clinical benefit [2], and the experience is often distressing and sometimes traumatising for service users and carers, contravening otherwise valued principles of consent and collaborative decision making [7,8]. It may undermine therapeutic relationships, engagement with services and overall mental health outcomes [9,10]. Inpatient admissions are also costly, especially detentions, which are typically longer than voluntary admissions. In England, inpatient care is estimated to use over a third of the whole mental health care budget [11].

People from Black ethnic groups are at increased risk of being detained in hospital across high-income, majority White population countries [12]. In England, they are four times more likely to be detained than their White British counterparts [6]. This disparity has been attributed to a combination of socio-economic factors, cultural misunderstandings, and systemic biases [13]; however, these explanations lack robust empirical support [12]. Data from the UK and other European countries also indicates a greater risk of detention for males than females [14–16].

A review of recent studies worldwide found the highest risk factors for being detained were previous compulsory admissions and a diagnosis of psychosis [15]. This review also found that medication non-adherence, police involvement during the process of detention, being unemployed, single, and receiving welfare benefits were also linked with increased risk of detention. Other research found socio-economic deprivation [15,17], having a risk of aggression [18], and lack of community services preceding admission [16–18], suicidal ideation and being admitted outside of normal service hours [19,20] to also be significantly associated with detention.

Qualitative findings on experiences and views about detention have potential to supplement the gradually growing quantitative body of evidence on risk factors for detention by illuminating individual pathways and contextual factors leading to detention, and by generating novel ideas for strategies for preventing detention. The views of clinicians are also of considerable interest, as they may provide a useful perspective on key factors contributing to detention and on any potential opportunities to avert this maximally coercive treatment at any stage [21,22]. One qualitative study conducted in Norway investigated staff and service user perspectives on pathways towards a referral for compulsory detention [22]. These included patient-level issues such as treatment cessation, aggression, substance abuse, housing, and employment issues, limited social networks and service-level factors such as lack of collaboration between services, of crisis plans, and of individualisation of care. A mixed methods study carried out in Inner London explored the views of clinicians involved in assessments for compulsory detentions about potential approaches to reducing detentions [23]. These included: having higher detention thresholds; reducing delays in MHA assessments; facilitating informed decision-making, whereby service users' broader support network and their background are more thoroughly considered; having better resourced community services offering early detection and intervention and alternatives to detention.

Beyond these studies, little qualitative research has so far focused on clinician perspectives on pathways to detention and ways of intervening to prevent this. Findings from such research has potential to inform the development of interventions to prevent detention. We therefore included an investigation of clinician views on this topic in the initial intervention development of the FINCH study [24], a National Institute for Health and Social Care Research (NIHR) (reference: NIHR201739). FINCH aims to adapt an intervention, previously validated in Switzerland, to reduce compulsory detentions in the UK. In this co-produced component of the FINCH study, we aimed to explore the views of a varied group of clinicians who work with people who are detained on factors that lead individuals to be compulsorily detained under civil sections of national mental health legislation and potential pathways to prevent these detentions. A study exploring this topic from a service user perspective was also conducted and is reported in a separate paper.

## Methods

### Ethics statement

Ethical approval was obtained for this study from University College London (UCL) Research Ethics Committee (Ethics ID Number: 15249/002) for the first wave of recruitment and data collection through non-NHS routes. For the second wave of recruitment in the NHS, Health Research Authority (HRA) and NHS Research Ethics Committee (REC) approval was granted by the London - Bromley Research Ethics Committee (IRAS: 300671; Protocol number: 143180; REC reference: 21/LO/0734).

### Research team

Academic researchers at UCL led the study's design, conduct, and analysis in collaboration with the FINCH co-production team. The co-production team included lived experience researchers (LERs). These individuals brought invaluable first-hand knowledge of being detained under the MHA or supporting close family or friends through such experiences. They participated actively in all aspects of the research process, ensuring that their lived experiences were central to the study's development. The team also included clinicians and academic researchers with diverse professional qualifications, including PhDs, medical, clinical psychology, social work and occupational therapy qualifications. Some of our academic researchers are practicing clinicians involved in decisions to detain (SJ, LB, TBH) or involved in providing care for detained service users in hospital (LW, FL).

The research team was intentionally diverse, representing a broad range of ethnicities (including 10 members from ethnic minority groups), ages, genders (defined as the socially constructed characteristics of women, men, or other gender identities), and geographic regions across the UK, including Lancashire, Birmingham, Manchester, Brighton, and London, and rural as well as urban areas.

The topic guide was developed collaboratively with all the team members, and although not subject to separate pilot testing phase, it was refined through extensive discussions and iterations. The interviews were conducted by several academic researchers (AK, MB, CHH) and one clinician (LB) with extensive qualitative expertise. The entire research team actively participated in the analysis process. For team members with limited qualitative experience, tailored training in qualitative analysis was provided to ensure their meaningful contribution.

Our multidisciplinary research team minimised the dominance of any single perspective. For members of the co-production group with lived experience, a monthly reflective space provided emotional support and a forum to discuss the research process and its emotional impact with core members of the UCL research team. Additionally, during regular team meetings throughout the analysis process, the team engaged in ongoing reflexivity, critically examining our biases, roles, and the power dynamics that could influence the research.

## Participants

Clinicians who currently are employed in an NHS mental health service or local authority and work with service users who were detained within the last 6 months under section 2 or 3 of the MHA were eligible to take part in the study. Section 2 or 3 of the MHA allow for people who are deemed to have a mental disorder to be detained in a psychiatric hospital because this is considered necessary for their own health or safety or protection of others to be detained in a psychiatric hospital. Section 2 allows a 28-day detention for assessment and Section 3 a renewable 6-month detention period for treatment [25].

Purposive sampling was used to ensure clinicians represented a variety of professional roles and settings across England that are directly and indirectly involved with the MHA assessment and with working with currently or recently detained patients (including community staff working with them pre-admission and post-discharge from compulsory admissions. We reviewed our sample for representativeness of main professional groups and clinician demographics during recruitment and carried out a second wave to include those with perspectives or roles not well represented.

## Recruitment

Recruitment occurred in two phases with clinicians who worked in acute inpatient and community mental health services and local authorities across England. For the first phase of interviews (September 2021 to January 2022), study advertisements were circulated by a range of Twitter accounts linked to the study, its home institution (UCL) and the research team and within networks of professionals linked to university departments. To recruit participants from professional groups under-represented from our initial social media recruitment, a second wave of recruitment (January 2023 to February 2023) was completed at one London NHS Trust.

## Data collection procedure

The study utilised a qualitative design, using semi-structured interviews. Questions were asked regarding clinicians' recent experiences of working with detained patients, and their views about what factors may have led to that detention and by what means detentions could have been prevented. The guide also included specific exploration of views about reasons for people from Black or Black British backgrounds being more likely to be detained, and potential ways to reduce the risk of future detentions. Please see S1 Text for the full topic guide. Previous study findings informed the guide, which was iteratively reviewed and refined with the co-production group.

Interviews were conducted by academic researchers (MB, AK, CHH, LB) via Microsoft Teams or Zoom. There was no prior relationship between interviewers and participants before the study. At the start of each interview, interviewers provided a brief introduction but did not disclose their personal goals, motivations for the research, or additional personal details. Verbal Consent to take part in the interviews was recorded prior to beginning each interview. Socio-demographic information was collected via a secure online survey on Opinio (a secure university online survey platform). The interviews were video recorded, although participants could opt to have their camera off, and lasted between 30–60 minutes in duration.

Audio recordings of the interviews were transcribed verbatim by a professional transcription company, with which UCL has a data sharing and privacy agreement.

## Data analysis

Data were analysed using template analysis [26] a form of thematic analysis [27]. This involves identifying and organising themes using a coding template that are based on a subset of transcripts. These are then applied to additional transcripts, and adapted and refined as further data is analysed. This approach allows for analysis by a group of researchers focusing on collectively defining meanings and structure of themes during the analysis process.

Drawing on a small sub-sample of transcripts, six researchers (AK, MB, JP, TCB, LB, CHH) were initially given 1–2 transcripts each and were invited to familiarise and code transcripts and propose a basic set of themes to help organise data and develop a preliminary coding framework. The entire team subsequently reviewed and discussed the preliminary framework, providing suggestions for a revised draft of the framework for the analysis team to line-by-line code against. An iterative process was employed in which each team member read and coded at least one transcript, followed by regular team meetings to critically examine the framework, refine existing themes, and propose new themes. This collaborative collective approach to analysis allowed for higher validity with increased reflexivity, in addition to allowing the perspectives of service users, carers, and clinicians to inform the analysis.

## Results

A total of 23 participants took part in the study. Nine (39.1%) were male, with the largest group being White British or Irish (47.5%), followed by Asian/Asian British/Indian (26.1%). Psychiatrists were the largest professional group (43.5%), and the most frequent work setting was inpatient and crisis services, 60.9%. Over half of the sample were based in London (65.2%). Full participant characteristics can be found in Table 1.

### Thematic analysis

Three major themes were identified regarding clinician views, (a) service user factors that increase risk of compulsory detention, (b) aspects of services that contribute to compulsory detentions and (c) potential pathways to reducing compulsory detention. Each theme contained several sub-themes. The themes outlined in Table 2 are explained further below and exemplified using participant quotes. Further quotes relating to each theme and subtheme can be found in S1 Table.

### Service User Level Factors Influencing Risk of Being Compulsorily Detained

The first theme encompasses factors related to service users that participants identified as contributing to compulsory detentions:

#### High Levels of Risk

Most participants discussed patients posing a risk of harm to themselves or to others as influencing detention. If high levels of risk were perceived as present and could not be effectively managed in the community, it was likely that compulsory detention would result:

*"He had stopped working and he had been, at times, reacting in distress to ongoing voices, had left his [hostel] accommodation without clothes in the middle of the night to run into the street because voices suggested he should do that."* [P15]

#### Previous/underlying Trauma

Traumas were also identified as contributing to people being compulsorily detained. This included previous experience of traumatic life events:

**Table 1. Sample Characteristics.**

| Gender | Male | 9 (39.1%) |
|---|---|---|
| | Female | 14 (60.9%) |
| **Ethnicity** | White British or Irish | 11 (47.8%) |
| | Asian/Asian British/Indian | 6 (26.1%) |
| | Black British/ Black African or Caribbean | 2 (8.7%) |
| | Any Other White Background | 4 (17.4%) |
| **Job Title** | Consultant Psychiatrist/Psychiatrist | 10 (43.5%) |
| | Mental Health Nurse | 3 (13%) |
| | Service manager/Team manager | 2 (8.7%) |
| | Clinical Psychologist | 2 (8.7%) |
| | Trainee Clinical Associate Psychologist | 1 (4.35%) |
| | Approved Mental Health Professional (AMHP) | 4 (17.4%) |
| | Occupational therapist | 1 (4.35%) |
| **Place of work** | Acute mental health services (inpatient and crisis services) | 14 (60.9%) |
| | Community mental health services | 3 (13%) |
| | Both | 5 (21.75%) |
| | Unknown | 1 (4.35%) |
| **Region** | London | 15 (65.2%) |
| | East of England | 2 (8.7%) |
| | South West England | 2 (8.7%) |
| | North West England | 1 (4.35%) |
| | South East England | 2 (8.7%) |
| | Yorkshire and Humber | 1 (4.35%) |

*"It was this sense that it was a sensitive period of time for that reason that had escalated the extreme distress that she can feel, and escalated, you know, and that as a result escalated the risk to herself."* [P4]

Personal experiences of racism or its intergenerational effects were also highlighted as contributing to vulnerability to poor mental health and thus risk of detention:

*"you've got people in black communities who have experienced terrible trauma either in their own life or in their parent's generation through migration you get here and then you get treated really badly. So it's hardly surprising you break-down."* [P6]

Other specific traumas that may affect service users included experiencing challenging childhood experiences such as being in the foster system, child bereavement, loss of custody of children, and trauma linked to low socioeconomic status.

## Service users' perceptions of their own mental health

The concept of insight was mentioned by some participants, who reported that lack of capacity and decision-making, lack of awareness of the nature of illness experiences and unwillingness to engage in treatment can lead to service users being detained. In these cases, treatment against the person's will was seen as the only option as the service user would not engage in treatment or agree to an informal admission:

**Table 2. Overview of Themes and Sub-Themes.**

| Main Theme | Sub-Themes |
| --- | --- |
| Service user level factors influencing risk of being compulsorily detained | • High levels of risk<br>• Previous/underlying trauma<br>• Service user perceptions of their own mental wellbeing and/or diagnosis<br>• Non-adherence to medication<br>• Lack of support from the service user's family and networks<br>• Disadvantage and discrimination<br>• Cultural attitudes to mental illness and services |
| Service/clinician level factors contributing to people being compulsorily detained | • Lack of communication and continuity in care<br>• Historical lack of trust<br>• Clinician biases and assumptions<br>• Lack of resources and disruptions in care<br>• Lack of variety and choice of treatments/care offered leading to compulsory detention<br>• Systemic and institutional barriers to engagement |
| Potential pathways to reducing compulsory detention | • Improving quality of care, continuity of care, and communication<br>• Increasing access to patient & family level intervention<br>• Investing in services<br>• Choice regarding medication<br>• Offering variety of treatments/care and alternatives to compulsory detention<br>• Improved discharge planning |

*"She was quite manic, quite paranoid, difficult to interrupt, lacked insight. All these terminologies really, but basically, she was quite adamant that she wasn't unwell, that she shouldn't have been brought into hospital on the first occasion, that she was safe, in order to go home."* [P5]

**Non-adherence to medication**

Not taking prescribed medications was reported to be a common factor leading to detentions. *"And a lot of the people that I see who get re-admitted, a factor is probably discontinuation of drug treatment, and which may lead to, like, a what's seen traditionally as a relapse in a person's condition, and that reinforces the notion that they need drugs."* [P4]

Participants identified several reasons for medication non-adherence, including doses perceived as being too low to be effective, distrust of psychiatric medication, and participants discontinuing use once they felt better, believing medication continuation was not necessary:

*"We do see frequently that people relapse because they stop taking their medication. And that may be because they've got better, they feel better, so they don't think they need the medication anymore."* [P16]

One participant noted that service users from Black backgrounds are more often diagnosed with schizophrenia, leading to higher rates of depot antipsychotic prescriptions. The visibility of non-adherence to these treatments was said to increase the likelihood of detention, contributing to their over-representation in detentions.:

*"If you think about treatment, you're [referring to Black service users] more likely to be on a depot antipsychotic rather than oral antipsychotics […] We often end up down the route of an MHA assessment when people have stopped taking medication. Now, if you stop a depot, we know about it instantly because we give it to you. If you're a black man receiving a depot, you're more like to be on a depot, and you stop it, we know."* [P17]

## Lack of support from the service user's family and networks

A lack of social support, particularly from family, was highlighted as a factor contributing to repeat detentions among service users. This was largely due to service users choosing to be in isolation, not living close by to family or having no relationship with their family when they are in the community:

*"Well, his friends and family lived elsewhere, so in a different city, so that was part of the difficulty. So, he didn't have much in the way of social structure around him."* [P16]

Limited social support in the community leads to the service user's mental health to deteriorate to the point of crisis, because no one is around to provide support sooner:

*"So, definitely, sometimes people who have less support around them can be a bit more tricky to be in touch with. You might find out, only, a lot later things have progressed a lot more."* [P15]

One clinician stated that a lack of understanding among family members regarding the service user's mental illness can act as a stressor contributing to detention:

*"One of the stressors is the person's relationship with their family or their family's understanding of their problems. They may feel that they're very pressured by their family to do things that they actually can't quite manage yet, because they're not well enough and their family are perhaps expecting too much of them. Sometimes, the family are highly critical, but yes, also, family can be the opposite."* [P16]

## Disadvantage and discrimination

Participants reported various social adversities that may contribute to deterioration of mental health and to a lack of help-seeking. These included socioeconomic status, racism, poor housing, unemployment, and lack of social support:

*"Social factors like poverty, like unemployment, like racism and stigma, social exclusion of all forms, all play a part in a person perhaps being more likely then to have some mental health problems if they've experienced that, kind of, adversity, that, kind of, social trauma."* [P4]

Some expressed helplessness that these social adversities are powerful influences that they could do relatively little about:

*"Benefits and asylum status, for example…. It feels really difficult to support them because we don't have any power to do anything about those things and so we can't sort out the basics for our patients."* [P17]

## Cultural attitudes to mental illness and services

Participants spoke of the influence of cultural factors on how people understand their problems and whether they seek help was also reported.

One clinician observed that, in certain cultures, mental illness is often viewed as taboo and not recognised as a medical condition. Instead, it may be interpreted through a spiritual or supernatural perspective:

*"In some cultures, mental illness is seen as a taboo, it's seen as this is maybe the devil, black magic. If somebody in your family has got mental illness, then the stigma from the cultural aspect as well is a big thing, as well as wider society."* [P9]

This cultural interpretation of mental illness was seen as resulting in individuals turning to alternative forms of support, rather than seeking conventional medical treatment:

"*So, they might be seeking maybe more spiritual help, rather than a combination of both.*" [P9]

A further clinician also noted in certain cultures and communities, individuals may be reluctant to seek external support, preferring to care for their family members at home:

"*The Asian community tends to want to look after their family members at home within the family environment. And that can be a challenge in terms of delivering an effective treatment in a timely fashion. So that can be a challenge in terms of a barrier to providing a treatment […] Those types of communities are not keen for Mental Health Act Assessments or their clients to be in hospital.*" [P3]

### Service/clinician level factors contributing to people being compulsorily detained

The second main theme encompassed key systemic, structural, and clinician-level factors that influence detention practices.

### Lack of communication and continuity of care

A lack of clear communication and joined up working across the service user's care pathway was identified as a risk factor for being detained. Participants explained communication with service users about their care can be poor, and that services often do not communicate or collaborate well with each other, resulting in care plan not being followed through and service user needs not being met:

"*…inpatient and community teams, it's very disjointed…because everyone has such a big caseload, the community team isn't even aware that a person was admitted… we've had cases where people have been on a depot, but the community team wasn't told that they were getting discharged, so they didn't know they had to follow up this depot.*" [P20]

### Historical lack of trust

Numerous participants viewed service users not trusting mental health services as a key contributor to compulsory detentions. Various reasons were suggested for this, including previous experience of poor care and police involvement in treatment:

"*Patients who still remember and are still traumatised by police officers crawling through windows to gain access to their properties to detain them*". [P19]

"*I think there wasn't enough understanding and communication between her and her community team. I think that, as she became increasingly suspicious of them, and they became, in her mind, increasingly attempting to control and police her, and with not a lot of collaborative relationship building and trust formation.*" [P7]

### Clinicians' biases and assumptions

Clinician biases and preconceptions were also seen as contributing to an increased likelihood of detaining certain service users. Many participants stated that if a service user has been detained previously, it is assumed that detention is likely to be needed again in future:

*"I think, often, if that's happened before, we sometimes fall into the trap that it's a foregone conclusion, that's what's going to happen again rather than perhaps reappraising the situation and thinking about what an admission had achieved and thinking about whether there are alternatives."* [P17]

Participants mentioned other biases such as age and gender:

*"I certainly suspect that we have a higher threshold of detaining elderly women than young men."* [P6]

Another form of bias was the higher likelihood of individuals with psychosis being detained compared to those with depression:

*"Someone with psychosis is more likely to be admitted under the Mental Health Act than someone with depression."* [P18]

Most participants felt that racial biases and stereotypes influenced likelihood of compulsorily detention, with some ethnic groups tending to be perceived as inherently a higher risk and thus more likely to require compulsory detention:

*"The way that risk is perceived is different, in my view, sometimes, depending on what your ethnicity and your cultural background is."* [P19]

This was reported as particularly true for service users from Black backgrounds:

*"I mean, society, as such, is essentially racist […] It's not an intentional attitude, I guess, it's a fear response associated with it. So, I think males, Black males are seen as more dangerous, and more risky. And the reason for that has to do with the colour of their skin, largely. In a profession, in a setting that the people who make the decisions are largely of a different race, and different colour."* [P23]

One participant stated that clinicians need to be better at reviewing whether detention is necessary:

*"It becomes the habit or routine response of the service to a crisis, rather than to rethink or review their part in the process that led to them being re-sectioned."* [P1]

## Lack of resources and disruptions

All participants spoke about compulsory detentions being linked to a lack of staffing, overstretched services, fragmented service provision and long waiting lists across inpatient and community services. Participants spoke about how overstretched community services meant that service users weren't receiving adequate care to maintain stability and facilitate recovery, leading to relapse, acute crisis and ultimately being detained. Participants also described hospital discharges as often rushed and driven by bed pressures, rather than timing being in the best interest of the service user:

*"And, as we know, services have been stripped out by years of austerity, so community services have been, you know, a victim of that, as have lots of other services like the police and so forth, that we really are seeing the effects of austerity now in terms of the resources within community and preventative services, that the staffing levels, recruitment, and retention is really problematic."* [P4]

*"There are certain people that get sectioned again because the amounts of resources in the community are not there enough to provide a level of intensive support in the community."* [P22]

### Lack of variety of treatments/care offered leading to compulsory detention

Participants spoke about how treatment mainly relied on pharmacological treatments due to the limited availability of psychosocial interventions for people in both inpatient and community settings:

> *"People, you know, are leaving hospital without the psychological skills that they need to be able to manage this if it were to happen in the community."* [P12]

Inpatient care was described as particularly limited, with the focus being too narrowly on stabilising and discharging rather than providing a holistic care package to facilitate personal recovery:

> *"I would say that the hospital admission process and the stay itself is, kind of, geared to the discharge itself and it's a, kind of, machine that's made and its main objective is discharging people from it."* [P10]

### Systemic and institutional barriers to engagement

Participants spoke about how the current mental health care systems excluded marginalised groups, leading to poor access to care, and worsening of mental health issues to a point of a crisis resulting in detention. Some participants said that mental health services are biased in relation to race. The people delivering services do not reflect the ethnic backgrounds of the services users or may not be culturally competent, so there is a lack of cultural understanding:

> *"A big discussion around the system being racist in terms of the approach, again, I think if you think about simple things like a delusion is by definition a fixed belief outside of the cultural norm. I mean, how on Earth can I be claiming, working in [inner London location] that I know everyone's cultural norm? My cultural norm may look like a delusion for you, so it's not so simple to kind of ignore these factors."* [P15]

Also, some participants believed that the potential barriers for service users from ethnic minority backgrounds, such as speaking a different language or having different beliefs about risk are not considered when developing and providing mental health services:

> *"The institutional response … I suspect that certain ethnic groups, possibly black men, … might have access issues to services, services might not be very visible to them. They might not know how to access them, or they might find barriers when they do try to access them."* [P4]

### Potential pathways to reducing compulsory detention

This theme relates to how participants thought improvements could be made to current practice to reduce the likelihood of a service user being compulsorily detained.

### Improving quality of care, continuity and communication

The need for more collaborative care and joined up working between service users, inpatient and community teams was a prominent suggestion. This was especially important when a service user was in crisis. Collaborative crisis planning, involving key stakeholders, was highlighted as essential:

*"Well, I have seen care plans that have been used brilliantly and involve collaboration with the person to understand what their needs are, think what they would like to see happening in terms of plans, think about what they would like to see happening in terms of crisis, having this plan done together […] I've seen that less often, compared to the times I've seen the care plan being a bit of a tick box exercise."* [P15]

Having a meaningful therapeutic relationship with key workers who know service users well and can offer continuity of care was emphasised as key to preventing detentions as they can spot early warning signs, tailor treatment, and prevent the worsening of symptoms:

*"If you've got a really good relationship with your care coordinator, it could reduce the chance of you being detained because that care coordinator is able to trust in the relationship to know when it's absolutely necessary."* [P6]

### Increasing access to a wider range of service user & family interventions

Improved access to preventative and inclusive psychosocial family interventions, such as Open Dialogue, which foster meaningful and collaborative relationships and bring everyone together, was reported to reduce the need for detention:

*"There've been approaches, like Open Dialogue, that advocate a lot more strongly, both, for the inclusion of the important people the person wants to bring in and also a more collaborative flat hierarchy. I appreciate sometimes our system is a little bit hierarchical in terms of power dynamic, so it's not just about bringing people on but trying to work together."* [P15]

Family and carer involvement at a time of crisis was seen as particularly important as they may be best placed to support the service user during a difficult period:

*"All of the community teams, crisis teams, inpatient units have to involve family and carers because they're the people that are also going to help support, keep this person well."* [P22]

### Investing in services

Multiple participants said if the community teams had more resources, they could review service users more regularly, build stronger relationships, and provide more intensive support. With time to build trust between services users and clinicians, relapse prevention work can happen and reduce future chance of being compulsorily admitted:

*"If you had really well-staffed teams, you would have lots of time to spend with people who have been in hospital to do that important relapse prevention stuff. And that would be the best way to potentially avert future admissions under the MHA."* [P19]

### Choice regarding medication

It was highlighted that choice and collaboration regarding medication was important to ensure medication was acceptable to service users. As medication non-adherence was often a key factor in relapse and ultimately detention, tackling the factors that may cause someone to stop taking their medication was seen as important.
One participant talked about formulating and intervening with the reasons for medication non-adherence:

                                    

*"Medication is one thing, but if it is about somebody not wanting to take medication, then of course there needs to be a conversation about why that is and how can we change the medication, or use a different formulation that makes it easier for the person to have it? Is it that the side effects are unbearable or is it simply that the person just doesn't want to take medication because they don't accept that they're unwell, in which case".* [P16]

**Offering variety of treatments/care and alternatives to compulsory detention**

Increased availability of a wide range of community services offering both crisis prevention and a range of approaches to crisis intervention were thought to be potentially helpful in preventing detentions, including day services, crisis houses and more responsive substance misuse services. Services users could be supported and treated when they are less unwell and have more insight, which could prevent someone from needing to be compulsorily admitted:

*"If we can get to people at an earlier stage … restructure and repurpose and fund properly community services to have … localised earlier, more preventative impact."* [P4]

**Improved discharge planning**

Participants thought that many service users are discharged from inpatient units before they are well due to operational issues such as bed shortages. Participants felt that clinically appropriate discharge with an agreed, well-conceived, communicated plan for the service user, community team and support network is needed to prevent readmission:

*"…to discharge someone quickly and rescind the section quickly…person comes back and is re-detained […] making sure someone is well enough at the point of discharge, with a smooth- with proper support arranged from a community team."* [P18]

## Discussion

### Principal findings

Three overarching themes were identified, encompassing clinicians' perspectives on factors that may lead to compulsory detention and suggestions to reduce it. The first theme, *"Service user level factors influencing risk of being compulsorily detained,"* reports perceived patient-level factors including presenting with high levels of risk, non-adherence to medication, lack of insight into their own mental health and/or diagnosis, experiencing previous trauma, having unstable family and wider social networks, and social and political factors such as race, culture, and employment status. The second theme, *"Service/clinician level factors contributing to people being compulsorily detained,"* describes service-level factors contributing to detention and areas for improvement to reduce future detentions. These include lack of communication and continuity of care, lack of resources and service disruptions, and lack of variety in treatments and care offered in the community. A historical lack of trust between services and patients, clinician biases and assumptions towards service users, and systemic and institutional biases to engagement were also noted as contributing factors. The final theme, *"Potential pathways to reducing compulsory detention,"* delineates possible suggestions to reduce detentions, including improving discharge planning, offering choice in medications, investing in services, and increasing access to patient and family level interventions.

### Findings in context to other literature

Various patient-, service-, and context-level factors contributing to detention have been identified in the current study. At the patient level, previous qualitative research highlights patients deciding to stop treatment and social adversities as

key contributors to referrals for compulsory detention [22]. Our findings align with this, with clinicians identifying medication nonadherence, as well as effects of social adversities—such as low socioeconomic status, racism, poor housing, unemployment, and lack of social support—as factors contributing to deteriorating mental health and repeated detentions. These factors were also supported in a recent meta-analysis and narrative synthesis [15], which also highlighted additional factors such as lack of patient insight as strongly associated with compulsory detention. Some clinicians in our study noted that a lack of patient insight increased the likelihood of detention, as service users often refused treatment or informal admission.

At the service-level, numerous factors contributing to compulsory detention were identified in this study. These included poor communication and collaboration between services, leading to unmet service user needs and unfulfilled care plans. These findings align with previous qualitative research, which also identified insufficient service continuity and collaboration as a key factor in pathways to detention [22]. Additionally, clinicians in our study described how overstretched community services often left service users without adequate support to maintain stability and achieve recovery, resulting in relapse, acute crises, and eventual detention. This aligns with previous research, which suggested that reductions in the availability or quality of community mental health services may have contributed to the increase in detentions [2,18].

At clinician-level, our participants emphasised the role of clinicians' assumptions and biases clinicians in increasing the likelihood of detention. Previous research has identified a prior detention as one of the strongest predictors of future compulsory detention [15]. This aligns with our study's findings, where many participants observed that a history of detention often leads to a routine assumption that detention is necessary during a crisis, without considering alternatives. Thus, when conducting an MHA assessment, an in-depth exploration on individuals' situations and their contexts rather than making assumptions based on their historical presentations may help reduce detentions but requires additional time and resources.

Participants in our study also noted that clinicians may hold unconscious biases or assumptions related to age and gender, with detentions being more likely for younger males. This aligns with previous research, which has found that males are more likely to be detained [14,15] a finding that warrants further exploration.

Clinicians in the current study also explicitly highlighted a bias towards detaining Black men. Clinician biases and racial profile have been widely proposed as explanations for the over-representation of Black men in detentions [12], and there are recent efforts to improve clinical practice and introduce unconscious bias training in mental health care [28]. A major aim in recent policy in England and Wales [29] is to monitor ethnic inequalities and introduce initiatives to reduce discrimination in the implementation of Act, for example through training initiatives.

The current study identifies potential pathways in reducing compulsory detentions from the clinician's perspective. These include a need for collaborative crisis planning between clinician and service user, particularly prior to discharge [24]. This is in keeping with policy in England and Wales, in which the focus is on delivering personalised care and treatment plans, and with the limited available evidence on what can prevent detention, in which the best evidence so far is on forms of collaborative care planning [30]. Our study also highlighted the need for greater availability of community services to help divert service users from detention. This aligns with previous research suggesting that better-resourced, individualised community services, at least from clinicians' perspectives, may reduce detentions [21,23]. Additionally, specialist early intervention in psychosis teams have been shown to lower detention rates compared to generic community teams [31].

## Strengths of the study

The current study had several strengths including addressing a gap in the literature with limited qualitative evidence in this area. The clinicians in the current study have worked with many service users who have been detained and a majority had been involved in the detention process: thus, they have substantial direct experience relevant to understanding the process, although clearly views were subjective.

## Limitations of the study

A limitation of our study is the potential for interpretation bias, as clinicians' perspectives may be influenced by their personal experiences, professional roles, and institutional contexts. Despite our efforts to include a diverse sample of clinicians from various roles and settings involved in the MHA assessment, most participants were psychiatrists, which may limit the representativeness of the findings for other professional roles. Furthermore, as the majority of the participants were based in London, the findings may not fully reflect the views of clinicians working in other regions of England. The clinicians in this study were also predominantly White, which is a key consideration given the well-documented overrepresentation of Black individuals in compulsory detentions. This lack of diversity among clinicians may limit the study's ability to capture a full range of perspectives.

An additional limitation of our study is the reliance on qualitative data from clinician interviews, without supplementing these insights with quantitative data or other empirical evidence, which could have strengthened the findings and provided a more comprehensive understanding. However, a companion paper from the same research team complements this study by focusing on service user perspectives, exploring pathways to compulsory detention and strategies to prevent repeat detentions [32].

## Clinical and policy implications

Clinicians focused largely on deficiencies in current service organisation and delivery and potential ways of remedying these, suggesting a range of potential pathways to reducing compulsory admissions. Currently, staff see services in England as lacking in resources which leads to services being disrupted by factors such as high staff turnover and staff shortages. Staff also see as crucial: continuity of care, provision of a choice of interventions and services, and collaboration with services. It has previously been reported that rising detention rates over the past decade may reflect a reduction in the number of contacts per patient in community teams [2].

Over-hasty discharge without a clear care plan was seen as contributing to repeat crises and detentions, supporting a focus on high quality care planning in hospital [3,33]. This is consistent with previous quantitative research which found that, in England, shorter length of index inpatient admission is associated with greater risk of readmission [34].

Medication non-adherence was perceived as key contributor to detention, suggesting collaborative decision making to find best fit treatments as a potential avenue for reducing detentions [8].

An evidence synthesis of trials of interventions [1] which included detention as an outcome found evidence that some psychosocial interventions - joint crisis planning and Early Intervention on Psychosis teams – may help to reduce detentions: clinicians in our study were in agreement with a view that enriching community care and the psychosocial interventions offered has potential to avoid detentions.

Biases related especially to race but also to gender, age and clinical history were also widely seen as important drivers of detention decisions, suggesting a need to find effective ways of challenging clinicians' unconscious biases and assumptions, for example through cultural competence training.

## Research implications

Clinicians' views can generate rather than test hypotheses about how detention might be reduced: thus, further research examining relationships between factors such as continuity of care and the range of community services offered and compulsory detention rates would be highly helpful, as well as in depth examination of the making of compulsory detention decisions, and the potential role of clinician biases and of availability of admission alternatives. An international perspective on service system characteristics and decision-making processes in countries with different levels of compulsory detentions is also potentially helpful.

Future research could also compare the perspectives of clinicians and service users on risk factors for compulsory detention as the factors highlighted by both groups may constitute high priorities for targets for initiatives to reduce detentions. Further research also may be needed to understand any discrepancies in views between stakeholder groups.

## Supporting information

**S1 Text. Topic Guide.**
(DOCX)

**S1 Table. Supplementary Table of Quotes.**
(DOCX)

## Author contributions

**Conceptualization:** Fiona Lobban, Brynmor Lloyd-Evans, Sonia Johnson.

**Formal analysis:** Ariana Kular, Mary Birken, Lisa Wood, Jordan Parkinson, Theresa Bacarese-Hamilton, Louise Blakley, Chloe Hutchings-Hay, Patrick Nyikavaranda, Dilshard Alam, Raphael Ogbolu, Caroline Bendall, Lai Tang, Amy Nickson, Cathryn Revell, Henrietta Mbeah-Bankas, Lizzie Mitchell, Kathleen Lindsay Fraser, Valerie Christina White, Fiona Lobban, Brynmor Lloyd-Evans, Sonia Johnson.

**Investigation:** Ariana Kular.

**Methodology:** Ariana Kular, Mary Birken, Lisa Wood, Fiona Lobban, Brynmor Lloyd-Evans, Sonia Johnson.

**Writing – original draft:** Ariana Kular.

**Writing – review & editing:** Ariana Kular, Mary Birken, Lisa Wood, Jordan Parkinson, Theresa Bacarese-Hamilton, Louise Blakley, Chloe Hutchings-Hay, Patrick Nyikavaranda, Dilshard Alam, Raphael Ogbolu, Caroline Bendall, Lai Tang, Amy Nickson, Cathryn Revell, Henrietta Mbeah-Bankas, Lizzie Mitchell, Kathleen Lindsay Fraser, Valerie Christina White, Fiona Lobban, Brynmor Lloyd-Evans, Sonia Johnson.

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
