## [Decision Letter · Decision Letter 0]

PMEN-D-24-00442

Exploring Pathways to Compulsory Detention and Ways to Prevent Repeat Compulsory Detentions; Clinician Perspectives.

PLOS Mental Health

Dear Dr. Kular,

Thank you for submitting your manuscript to PLOS Mental Health. After careful consideration, we feel that it has merit but does not fully meet PLOS Mental Health’s publication criteria as it currently stands. Therefore, we invite you to submit a revised version of the manuscript that addresses the points raised during the review process.

We look forward to receiving your revised manuscript.

Kind regards,

Abid Rizvi

Academic Editor

PLOS Mental Health

Journal Requirements:

1. In the ethics statement in the Methods, you have specified that verbal consent was obtained. Please provide additional details regarding how this consent was documented and witnessed, and state whether this was approved by the IRB.

Additional Editor Comments (if provided):

Thank you for your submission to PLOS mental health. While the journal ensures speedy peer review. We were having trouble in getting the report from a second reviewer, which we just received. I am happy to inform you that your manuscript will be considered for publication after the following recommended corrections. In addition to the concerns and recommendation of the reviewers, authors are requested to address the following

Provide - Consolidated criteria for reporting qualitative research (COREQ ) checklist

While the manuscript aligns well with COREQ guidelines in many areas, the following aspects could be better addressed to fully meet the standards:

Include a statement about the researchers’ backgrounds, biases, and relationship with participants.

Explicitly describe the reflexivity process undertaken during data collection and analysis.

Provide more details on achieving data saturation.

The manuscript generally demonstrates good adherence to EQUATOR guidelines via COREQ but would benefit from addressing the gaps noted above.

Reviewers' comments:

Reviewer's Responses to Questions

**Comments to the Author**

1. Does this manuscript meet PLOS Mental Health’s publication criteria?

Reviewer #1: Yes

Reviewer #2: Yes

2. Has the statistical analysis been performed appropriately and rigorously?

Reviewer #1: N/A

Reviewer #2: N/A

3. Have the authors made all data underlying the findings in their manuscript fully available (please refer to the Data Availability Statement at the start of the manuscript PDF file)?

Reviewer #1: No

Reviewer #2: Yes

4. Is the manuscript presented in an intelligible fashion and written in standard English?

Reviewer #1: Yes

Reviewer #2: Yes

Reviewer #1: Thank you for giving me the opportunity to review the manuscript titled "Exploring Pathways to

Compulsory Detention and Ways to Prevent Repeat Compulsory Detentions; Clinician Perspectives"

Overall, it is a nice study and authors' rigor in conducting a qualitative research with a well described thematic analysis is commendable. The study offers significant insights into the factors influencing compulsory detention in psychiatric hospitals, highlighting both individual and systemic elements. Its findings underscore the potential for reducing involuntary admissions through improved resources, care quality, and addressing clinician biases. These insights lay a foundation for future research to explore targeted interventions and policies aimed at minimizing compulsory detentions, potentially transforming mental health care practices and policies. On the other hand, there are certain weak points too. Please see the "word file" for specific "reviewer comments". Incorporating the made suggestions or the highlighted issues throughout the manuscript will add to the validity and impact of this research. I will be happy to re-review the manuscript after the suggested changes.

Reviewer #2: I appreciate the opportunity to review the manuscript. It is good to see your effort in proving the known hypothesis about various high risk factors for detention of patients with psychiatric illness and recommendations to address these issues.

**Do you want your identity to be public for this peer review?** For information about this choice, including consent withdrawal, please see our Privacy Policy

Reviewer #1: No

Reviewer #2: No

---

## [Editor Report · Decision Letter 1]

Exploring Pathways to Compulsory Detention and Ways to Prevent Repeat Compulsory Detentions in England; Clinician Perspectives.

PMEN-D-24-00442R1

Dear Dr Kular,

We are pleased to inform you that your manuscript 'Exploring Pathways to Compulsory Detention and Ways to Prevent Repeat Compulsory Detentions in England; Clinician Perspectives.' has been provisionally accepted for publication in PLOS Mental Health.

Best regards,

Abid Rizvi

Academic Editor

PLOS Mental Health

authors have addressed all major concerns of the reviewers